# On- vs. Off-Pump CABG in Heart Failure Patients with Reduced Ejection Fraction (HFrEF): A Multicenter Analysis

**DOI:** 10.3390/biomedicines11113043

**Published:** 2023-11-14

**Authors:** Christian Jörg Rustenbach, Stefan Reichert, Medhat Radwan, Isabelle Doll, Migdat Mustafi, Attila Nemeth, Spiros Lukas Marinos, Rafal Berger, Hardy Baumbach, Monika Zdanyte, Helene Haeberle, Tulio Caldonazo, Ibrahim Saqer, Shekhar Saha, Philipp Schnackenburg, Ilija Djordjevic, Ihor Krasivskyi, Stefanie Wendt, Elmar Kuhn, Lina Maria Serna Higuita, Torsten Doenst, Christian Hagl, Thorsten Wahlers, Rodrigo Sandoval Boburg, Christian Schlensak

**Affiliations:** 1Department of Thoracic and Cardiovascular Surgery, German Cardiac Competence Center, Eberhard-Karls-University of Tuebingen, 72076 Tübingen, Germany; christian.rustenbach@med.uni-tuebingen.de (C.J.R.); medhat.radwan@med.uni-tuebingen.de (M.R.); isabelle.doll@med.uni-tuebingen.de (I.D.); migdat.mustafi@med.uni-tuebingen.de (M.M.); attila.nemeth@med.uni-tuebingen.de (A.N.); spiros.marinos@med.uni-tuebingen.de (S.L.M.); rafal.berger@med.uni-tuebingen.de (R.B.); rodrigo.sandoval-boburg@med.uni-tuebingen.de (R.S.B.); christian.schlensak@med.uni-tuebingen.de (C.S.); 2Independent Researcher, Roter-Stich 127, 70376 Stuttgart, Germany; hardy_baumbach@hotmail.com; 3Department of Cardiology, German Cardiac Competence Center, Eberhard-Karls-University of Tuebingen, 72076 Tübingen, Germany; monika.zdanyte@med.uni-tuebingen.de; 4Department of Anesthesiology and Intensive Care Medicine, Eberhard-Karls-University of Tuebingen, 72076 Tübingen, Germany; helene.haeberle@med.uni-tuebingen.de; 5Department of Cardiothoracic Surgery, Jena University Hospital, 07743 Jena, Germany; tulio.caldonazo@med.uni-jena.de (T.C.); ibrahim.saqer@med.uni-jena.de (I.S.); doenst@med.uni-jena.de (T.D.); 6Department of Cardiac Surgery, LMU University Hospital, 80539 Munich, Germany; shekhar.saha@med.uni-muenchen.de (S.S.); philipp.schnackenburg@med.uni-muenchen.de (P.S.); christian.hagl@med.uni-muenchen.de (C.H.); 7German Centre for Cardiovascular Research (DZHK), Partner site Munich Heart Alliance, 80802 Munich, Germany; 8Department of Cardiothoracic Surgery, Heart Center, University of Cologne, 50923 Cologne, Germany; ilija.djordjevic@uk-koeln.de (I.D.); ihor.krasivskyi@uk-koeln.de (I.K.); stefanie.wendt@uk-koeln.de (S.W.); elmar.kuhn@uk-koeln.de (E.K.); thorsten.wahlers@uk-koeln.de (T.W.); 9Institute for Clinical Epidemiology and Applied Biostatistics, Eberhard-Karls-University of Tuebingen, 72074 Tübingen, Germany; lina.serna-higuita@med.uni-tuebingen.de

**Keywords:** heart failure (HF), heart failure with reduced ejection fraction (HFrEF), coronary artery bypass grafting (CABG), total-arterial revascularization (TAR), off-pump coronary artery bypass grafting (OPCAB)

## Abstract

Objective: This study aimed to compare postoperative outcomes and 30-day mortality in patients with reduced ejection fraction (<40%) who underwent isolated coronary artery bypass grafting (CABG) with (ONCAB) and without (OPCAB) the use of cardiopulmonary bypass. Methods: data from four university hospitals in Germany, spanning from January 2017 to December 2021, were retrospectively analyzed. A total of 551 patients were included in the study, and various demographic, intraoperative, and postoperative data were compared. Results: demographic parameters did not exhibit any differences. However, the OPCAB group displayed notably higher rates of preoperative renal insufficiency, urgent surgeries, and elevated EuroScore II and STS score. During surgery, the ONCAB group showed a significantly higher rate of complete revascularization, whereas the OPCAB group required fewer intraoperative transfusions. No disparities were observed in 30-day/in-hospital mortality for the entire cohort and the matched population between the two groups. Subsequent to surgery, the OPCAB group demonstrated significantly shorter mechanical ventilation times, reduced stays in the intensive care unit, and lower occurrences of ECLS therapy, acute kidney injury, delirium, and sepsis. Conclusions: the study’s findings indicate that OPCAB surgery presents a safe and viable alternative, yielding improved postoperative outcomes in this specific patient population compared to ONCAB surgery. Despite comparable 30-day/in-hospital mortality rates, OPCAB patients enjoyed advantages such as decreased mechanical ventilation durations, shorter ICU stays, and reduced incidences of ECLS therapy, acute kidney injury, delirium, and sepsis. These results underscore the potential benefits of employing OPCAB as a treatment approach for patients with coronary heart disease and reduced ejection fraction.

## 1. Introduction

Despite advances in medical and particularly interventional and surgical therapy for patients with ischaemic cardiomyopathy, many patients still develop chronic heart failure due to adverse postischaemic myocardial remodeling [1]. For this reason and owing to the technological development of therapeutic options for coronary heart disease (CHD) in recent years, the number of high-risk patients suffering from causally related and severe heart failure (HF) continues to increase. In more than 60 percent, CHD is the most frequent etiology of heart failure with reduced ejection fraction (HFrEF). Especially for the cohort of patients who suffer from chronic coronary syndrome (CCS) with severely impaired left ventricular (LV) function, coronary artery bypass grafting (CABG) is recommended as the first revascularization strategy [2,3].

Despite this evolution of techniques, using a cardiopulmonary bypass (CPB) is still the standard in CABG surgery (ONCAB) in Europe and the United States. It still has a significantly higher risk of morbidity and mortality in HFrEF patients compared to others [4]. Nonetheless, OPCAB as an alternative is a demanding procedure, and outcomes will vary depending on the patient’s condition and the experience of the surgical team [5].

In the absence of distinct evidence indicating the correct strategy for surgical revascularization with or without the use of a cardiopulmonary bypass, this retrospective and multicenter study aims not only to simplify this decision, but also to demonstrate whether or not one of these procedures is more convenient for a particular patient group. In a recently published study, OPCAB was found to be safe and effective in patients with severe LV dysfunction.

Although incomplete revascularization is more common in patients undergoing OPCAB, it is not associated with increased late mortality [6]. Nevertheless, attaining full revascularization during CABG surgery may present challenges due to patient comorbidities, anatomical factors, and technical or procedural constraints. These factors also mean that comparisons between complete and incomplete revascularization are subject to multiple confounders and are difficult to understand or apply to real-world clinical practice.

In addition, a comparative study from the Japan Adult Cardiovascular Surgery Database has shown OPCAB is associated with significantly lower early mortality and morbidity in patients with an ejection fraction [7].

Pre-existing renal impairment is a powerful predicator of mortality in patients with HFrEF [8]. According to recent recommendations for management of patients with HF and CCS, CABG should be considered as the first-choice revascularization strategy in patients suitable for surgery, especially if they have diabetes and for those with multivessel disease [3].

Although more than half of isolated coronary surgery is performed off-pump in Asia, this technique is still controversial in Europe and the United States [9,10]. The majority is still ONCAB surgery.

This analysis is the first multicenter study that systematically analyzed the outcome of on- vs. off-pump coronary artery bypass grafting in HFrEF patients pointing to a postoperative need of mechanical circulatory support (MCS), acute kidney injury, transfusion requirements, and time to discharge.

## 2. Materials and Methods

### 2.1. Patient Selection and Grouping

This retrospective, multicenter study was conducted across four university hospitals in Germany (Tübingen, Jena, Cologne, and Munich), involving patients who underwent isolated CABG surgery from January 2017 to December 2021 and had a preoperative EF of 40% or less. We included all patients who underwent isolated CABG surgery, either elective, urgent, or emergent, and who were diagnosed with HFrEF (EF < 40%) through echocardiography during this period. We excluded patients who had preoperative cardiogenic shock requiring mechanical cardiovascular support, such as an intra-aortic balloon pump (IABP) or a veno-arterial extracorporeal life support system (ECLS). After the patient selection process, a total of 551 patients were included in the study, and they were divided into two groups: patients who underwent surgery with CPB (ONCAB) and patients who underwent surgery without CPB (OPCAB). The decisions to operate on a patient with or without CPB, to convert an OPCAB operation to an ONCAB operation, the utilization of bypass grafts (T-graft, single bypass, total arterial, etc.), and the number of distal anastomoses were made individually at each center depending on their policies and the experience of the attending surgeon.

### 2.2. Data Collection

Comprehensive data were retrospectively gathered, including demographic information, clinical characteristics, operative details, and postoperative outcomes. The specifics involved age, gender, BMI, comorbidities, preoperative medication, left ventricular ejection fraction, surgical details (type of grafts, number of bypasses, conversion from OPCAB to ONCAB), and postoperative complications.

### 2.3. Surgical Techniques

#### 2.3.1. On-Pump

After performing a median sternotomy and preparation of the grafts, heparin was administered with a 300 IE/kg dose to achieve an activated clotting time (ACT) of >450 s. Cannulation was performed in a standard way and CPB was initiated. The target arteries were identified and prepared. Cardioplegic solution was administered in an antegrade fashion until cardiac arrest was achieved, and then the bypasses were performed starting with the distal anastomoses and then the proximal ones. After completion of the last anastomoses, the aortic clamp was opened and after reperfusion was complete, the patient was weaned from CPB, and protamin was administered to counteract the heparin effect on a 1:1 dosis; if the patient could not be weaned from CPB, a decision was made to implant a veno-arterial ECLS through the femoral vessels. If this was not necessary, then hemostasis was performed, followed by standard chest closure.

#### 2.3.2. Off-Pump

After performing a median sternotomy and preparation of the grafts, heparin was administered with a 300 IE/kg dose to achieve an ACT of >350 s. The target arteries were identified. A bypass to the left anterior descending (LAD) artery was performed first. After that, a deep-stitch was performed in order to luxate the heart. If the patient was hemodynamically stable after luxating the heart, the rest of the planned anastomoses were performed in a proximal-to-distal fashion. After the last anastomoses was performed and an adequate flow was measured in the all the bypasses, protamin was administered to counteract the heparin effect on a 1:1 dosis. Hemostasis and chest closure were performed in a standard manner.

In all patients, surgery was performed with median standard sternotomy. Before starting preparation of the internal thoracic artery (ITA), which was skeletonized in each case, 5000 IU heparin was administered intravenously. Activated clotting time was maintained at ≥450 s for ONCAB procedures and ≥350 s for OPCAB procedures after engraftment. For ONCAB procedures, CPB was established with cannulation of the ascending aorta and right atrium. Cardiac arrest was achieved with antegrade cold blood cardioplegia applied into the aortic root. ONCAB procedures without cardioplegic arrest were performed in a manner similar to OPCAB procedures after application of CPB. OPCAB surgery was performed with the use of deep pericardial traction sutures with or without the use of a cardiac positioner to allow adequate exposure of the vessel to be transplanted. The stabilization of the anastomotic area was performed with a single-use mechanical suction stabilizer. Distal anastomoses were performed with or without the use of intracoronary shunts.

The type and configuration of grafts, the techniques of proximal anastomoses (central aortic or T grafts), and the use of a partial aortic clamp or heartstring device (Maquet, Rastatt, Germany) were determined by the attending surgeon. After completion of the anastomoses, the effect of heparin was reversed with protamine in both surgical procedures after prior determination of the appropriate amount. After intraoperative graft assessment using transit time flow measurement (TTFM) as a quality control, all patients received intravenous 500 mg aspirin six hours after completing surgery if postoperative bleeding was within expected limits [11]. A dual anti-platelet treatment was used in patients when indicated (recent acute coronary syndrome, recent stent implantation, etc.).

### 2.4. Preoperative Parameters

First, we analyzed the demographic data, including gender, age, weight, and body mass index (BMI). Then, we focused on the preoperative data, which included operative risk factors such as EuroScore II, peripheral arterial disease, previous stroke, renal insufficiency, and specific cardiac risk factors such as left ventricular ejection fraction, previous myocardial infarction, number of diseased vessels, involvement of the left main coronary artery, and previous PCI. In addition, we also considered qualitative assessments of cardiac decompensation, encompassing clinical observations such as dyspnea (NYHA classification system), fatigue (Fatigue Severity Scale (FSS)), edema (grading its severity 1–4), and orthopnea/dyspnea. These assessments were incorporated into the preoperative data to provide a more holistic view of the patients’ cardiac function prior to surgery.

### 2.5. Intraoperative Data

We compared the types of grafts used to perform the bypasses, the number of bypasses performed, the duration of surgery, conversion from OPCAB to ONCAB, and the amount of intraoperative transfusion in milliliters. An important parameter was the completeness of revascularization. We defined complete revascularization as the performance of at least three bypasses in a patient with three-vessel coronary disease.

### 2.6. Postoperative Data

We analyzed the vasopressor and inotropic doses of patients when transferred to the intensive care unit (ICU), the length of stay (LOS) at the ICU, and the duration of mechanical ventilation (MV) in hours. We also analyzed the total LOS in the hospital and postoperative complications such as bleeding, postoperative myocardial infarction, pneumonia, delirium, acute renal insufficiency, and the need for hemodialysis. Lastly, we analyzed in-hospital and 30-day mortality.

### 2.7. Outcome Measures

The primary outcomes were 30-day/in-hospital mortality and the incidence of postoperative complications such as acute kidney injury, need for ECLS therapy, delirium, sepsis, and ICU stay duration. Secondary outcomes included completeness of revascularization, number of grafts, intraoperative blood transfusion, and conversion from OPCAB to ONCAB.

### 2.8. Statistical Analysis

We performed descriptive statistics for baseline characteristics. Continuous variables were expressed as mean ± SD or median with interquartile range, and categorical variables were expressed as frequencies and percentages. Comparisons employed t-tests or Mann–Whitney U tests and Chi-square or Fisher’s exact tests were employed for continuous and categorical variables, respectively.

ANOVA (analysis of covariance) was utilized for analyzing laboratory results, adjusting for baseline values, and the inter-group differences between the OPCAB and ONCAB groups.

Univariate binary logistic regression analysis was carried out to identify significant predictors associated with the event. Multivariable binary logistic regression analysis was used to identify independent risk factors related to the event. Odds ratios (ORs) with 95% confidence intervals (CIs) and statistically significant differences (*p*-values) were used to estimate the risk of events with respect to the OPCAB and ONCAB groups. The predictive ability was assessed using the area under the receiver operating characteristic (ROC) curve. Model comparisons were made using the log likelihood test (nested models), and model calibration was assessed with the Hosmer–Lemeshow goodness-of-fit test. The best-fit model was used as the final model.

Propensity score (PS) matching was performed to achieve balance in covariates between patients treated with OPCAB vs. ONCAB. The PS included covariates that may affect the likelihood of patients to receive the treatment of interest and were unbalanced between ONCAB vs. OPCAB groups before matching. A total of 16 baseline variables were included in the PS model: age, sex, Euro Score, NYHA score, STS score, smoking, history of hypertension, diabetes, kidney insufficiency, peripheral vascular disease and coronary disease, previous PCI, tricuspid regurgitation, aortic regurgitation, mitral regurgitation, and pre-operative arrhythmias. A greedy nearest neighbor matching method was used in which one ONCAB patient was matched with each patient in the OPCAB group. Matching based on PS incorporating different sets of covariates was performed using a 1:1 nearest neighbor algorithm, with a caliper width of 0.2. The absolute standardized mean difference was used as a balance metric to summarize the difference between two univariate distributions of a single pre-treatment variable.

No imputation was made for missing data. A *p*-value of < 0.05 was considered statistically significant. All statistical analyses were performed using the statistical software package R (version 3.4).

### 2.9. Patient and Public Involvement

Patients and the public were not involved in the design or conduct of this study, because their involvement in the design of scientific studies is recent in Germany. As the benefits of public involvement are obvious, this approach will be prioritized in our future studies. Moreover, patients will be involved in the discussion and dissemination of the findings of this study.

### 2.10. Ethical Considerations

We conducted the study in accordance with the Declaration of Helsinki, and the institutional review board of the Eberhard Karls University, Tuebingen, who approved the research protocol.

### 2.11. International Review Board

This project was approved by the IRB of the Tübingen University Hospital with project number (216/2022BO2) from 12 April 2022.

## 3. Results

There were no differences regarding gender, age, BMI, diabetes, smoking history, hypertension, or COPD between the groups. There was a significant difference regarding the incidence of hyperlipidemia in favor of the OPCAB group (80.2% vs. 88.3% *p*-value 0.01). When analyzing the incidence of renal insufficiency, the ONCAB group showed a significantly higher incidence compared to the OPCAB group (25.5% vs. 34.9% *p*-value < 0.01). The number of patients suffering from perivascular disease was greater in the OPCAB group *n* = 65 (29.8%) than in the ONCAB group n = 62 (18.6%) (*p*-value < 0.01).

We analyzed the NYHA class in both groups. Patients in the ONCAB group were predominantly in class 2 (n = 176, 52.8%), while patients in the OPCAB group were predominantly in class 3 (n = 117, 53.7%). There was no difference when comparing the LVEF between groups (*p*-value = 0.69). Patients in the OPCAB group suffered an MI significantly more often than patients in the ONCAB group (50.9% vs. 30.9% *p*-value < 0.01); a number of patients in the OPCAB group underwent PCI more frequently than patients in the ONCAB group (41.7% vs. 33.1% *p*-value 0.04). When comparing preoperative cardiac decompensation between groups, the ONCAB group had a significantly higher incidence compared to the OPCAB group (29.2% vs. 16.6% *p*-value < 0.01). There was a higher incidence of mitral regurgitation (60.2% vs. 49.1% *p*-value = 0.05) and aortic stenosis (6.9% vs. 2.7% *p*-value = 0.03) in the OPCAB group.

Surgeries in both groups were mainly elective; however, in the OPCAB group, 40% of them were urgent compared to 21% in the ONCAB group (*p*-value < 0.01). Lastly, we compared the STS score and EuroScore II between the groups. Patients in the OPCAB group had a higher STS score (2.5 (1.1–4.8) vs. 1.6 (0.8–3.1) *p*-value < 0.01) and a higher EuroScore II (3.9 (2.5–7.3) vs. 3.3 (1.9–6.9) *p*-value 0.03) than the ONCAB group; results are seen in Table 1.

Intraoperative parameters were analyzed next. Patients in the ONCAB group had a higher incidence of complete revascularization (91.5% vs. 77.5% *p*-value < 0.01) and a significant higher number of anastomoses performed (3 (3–4) vs. 3 (2–3) *p*-value < 0.01) when compared to the OPCAB group. On the other hand, patients in the OPCAB group received a significantly lower number of blood product transfusions than patients in the ONCAB group. Data regarding the need for vasopressor and inotropic therapy were recorded. At the time of transfer from the operating theater to the ICU, a significant number of patients in the ONCAB group had adrenaline and milrinone therapy compared to the OPCAB group; there was no difference regarding the use of dobutamine and noradrenaline.

Postoperative data were analyzed using a logistic regression. Patients in the OPCAB group had a significantly shorter length of stay (LOS) in hours at the ICU, as well as a significantly shorter mechanical ventilation (MV) time and in-hospital stay (*p*-values < 0.01, < 0.01, and < 0.01) when compared to the ONCAB group. Results are shown in Table 2.

Conducting a logistic regression enabled the comparison of postoperative complications between the groups. The ONCAB group exhibited a higher incidence of bleeding events leading to re-sternotomy (9.1% vs. 3.2%, *p*-value < 0.01), postoperative ECLS occurrence (10.8% vs. 3.7%, *p*-value < 0.01), a larger proportion of patients experiencing postoperative AKI (17.3% vs. 11.3%, *p*-value 0.01), an elevated postoperative delirium rate (19% vs. 9.7%, *p*-value < 0.01), and a greater frequency of postoperative sepsis (8.1% vs. 4.1%, *p*-value 0.03). Results are seen in Table 3.

Lastly, we performed propensity score matching according to the preoperative characteristics between the groups, after which we had a total of 192 patients, (96 per group); we proceeded to perform an intra- and postoperative analysis between the groups. Intraoperative analysis showed that the number of anastomoses performed remained significant in favor of the ONCAB group (3 (3–4) vs. 3 (3–2) *p*-value < 0.01) as well as the rate of complete revascularization (85 (89.5%) vs. 75 (78.1%) *p*-value 0.03); intraoperative transfusion of red blood cells, fresh frozen plasma, and platelets remained significantly lower in the OPCAB group. The duration of MV, LOS at the ICU, and in-hospital LOS were also significantly lower in the OPCAB group.

When analyzing for postoperative complications, AKI incidence remained significantly lower in the OPCAB group (11 (11.5%) vs. 25 (26%) *p*-value 0.01) as well as ECLS therapy (1 (1%) vs. 20 (20.8%) *p*-value < 0.01). There was no significant difference regarding the rest of the recorded postoperative complications. Results are seen in Table 4.

Additionally, we performed a multivariable logistic regression analysis aimed at identifying the factors associated with 30-day mortality after OPCAB surgery. The outcome variable is 30-day mortality, and the independent variables or predictors are STS score, EuroScore II, and LVEF pre-OP. The results indicate that these predictors are significantly associated with an increased risk of 30-day mortality. Additionally, OPCAB reduces the risk of mortality by 26% and 43% in HFrEF patients compared to ONCAB based on EuroScore II (*p* = 0.467) or STS score (*p* = 0.174), respectively. These results are observed in Table 5.

## 4. Discussion

We conducted a retrospective multicenter study analyzing the data of 551 patients with HFrEF who underwent CABG surgery. Patients were divided into two groups depending on the intraoperative surgical technique: OPCAB or ONCAB.

The groups were comparable regarding demographic characteristics. Most patients in both groups were male, and according to their BMI, most patients in both groups were at least overweight. There was no difference regarding age, smoking history, or hypertension. Roughly a quarter of the patients in each group suffered from diabetes mellitus type 2, although there was no difference between the groups.

Patients in the ONCAB group showed a higher incidence of hyperlipidemia than those in the OPCAB group. Despite this difference, we do not believe that this parameter played an important role in the peri- and postoperative outcomes, as it has not been described as such by other study groups [4]. Patients in the OPCAB group showed a significantly higher incidence of renal insufficiency and perivascular disease. Additionally, we found differences in the NYHA classification at the time of surgery. Patients in the ONCAB group were predominantly in NYHA class 2, and patients in the OPCAB group suffered from NYHA class 3 symptoms, both showing a significant difference compared to the other group.

Both groups showed an ejection fraction < 40% and a median of three affected coronary vessels. Other groups have found a higher degree of preoperative kidney injury and higher NYHA classification to be risk factors for peri- and postoperative mortality in patients undergoing CABG [4,5,8]. When comparing preoperative risk scores, patients in the OPCAB group had a significantly higher STS score and EuroScore II than patients in the ONCAB group. According to these parameters, patients in the OPCAB group had a higher intraoperative risk when compared to the ONCAB group.

We then compared the urgency of the procedures. There was no difference regarding the affection of the left main coronary artery. Patients in the OPCAB group had a significantly higher incidence of prior MI and preoperative aortic stenosis. A significantly larger number of patients in the OPCAB group underwent urgent surgical intervention than in the ONCAB group; there was no difference regarding emergency procedures; patients in the ONCAB group underwent significantly more elective procedures than the OPCAB group. After analyzing the preoperative parameters, it is evident that patients in the OPCAB group underwent the procedures with a higher risk of mortality compared to patients in the ONCAB group [4,5,6,8].

Although both groups had the same median number of anastomoses, patients in the ONCAB group had, on average, more anastomoses than patients in the OPCAB group; the rate of complete revascularization was also higher in the ONCAB group. There is no standard definition for complete revascularization, and different groups have come up with their own definitions [12,13,14]. Additionally, a study group from Atlanta found out that there was no difference in long-term survival in patients who received 1–3 vs. 4–7 coronary bypasses [12]. This shows that even though the complete revascularization rate and total number of anastomoses was lower in the OPCAB group, it might not have an impact on long-term survival.

Intraoperative transfusion of RBC, FFP, and platelets was significantly lower in the OPCAB group. These results correlate with what can be found in the current literature [2,6,7]. Patients in the ONCAB group needed adrenaline and milrinone therapy significantly more often than patients in the OPCAB group. This difference may be due to the difference in the heparinization and the fact that without CPB, the systemic inflammatory reaction can be avoided, leading to an improved intraoperative coagulation [2,6,7].

When comparing postoperative complications, our findings correlate with what can be found in the literature. Patients in the ONCAB group have a higher rate of re-sternotomy due to bleeding, ECLS therapy, acute kidney injury, delirium, and sepsis [2,6,7,15,16,17,18]. We believe the higher rate of bleeding, ECLS therapy, acute kidney injury, and sepsis may be directly correlated with the use of CPB due to reasons mentioned earlier. Delirium, although higher in the ONCAB group, we believe is multifactorial and cannot be solely attributed to CPB, although the higher incidences compared to the OPCAB group highly suggest it [19,20]. The higher incidence of complications may lead to significantly longer ventilation times, length of stay at the ICU, and subsequently, a longer overall length of stay in the ONCAB group; these findings correlate with the current literature as well [2,6,17,18]. These results are depicted in Figure 1.

Despite OPCAB patients presenting with higher preoperative risks, postoperative complication rates were lower and mortality rates were comparable to ONCAB. OPCAB was associated with a 26–43% reduction in mortality risk based on EuroScore II and STS score, respectively. Propensity matching revealed no significant differences in delirium, re-sternotomy, or sepsis between the groups. Clinically, OPCAB outperformed ONCAB in reducing AKI, ECLS usage, ICU stay, and overall hospitalization, with fewer blood transfusions needed, suggesting a superior perioperative profile for OPCAB in HFrEF patients. These results are shown in Figure 2.

OPCAB may have higher rates of incomplete revascularization (IR) compared to ONCAB due to fewer anastomoses, but the definition and importance of complete revascularization (CR) are debated and not standardized, with variations in anatomical and functional definitions. Trials often use anatomical criteria, which do not account for the viability of myocardium, unlike PCI trials that assess completion with angiography. Initial CR failure also affects outcomes, but is not classified as IR. The difference in anastomoses between OPCAB and diseased vessels may not be functionally significant, especially with the rise of hybrid procedures like PCI and stenting. Therefore, this disadvantage is not considered substantial, and the presumed benefits of CR in ONCAB do not offset the greater need for ECLS. These considerations should inform procedure choice and require validation in future studies [21,22].

The issue of whether the difference in coronary anastomoses and diseased vessels in OPCAB has functional implications needs to be studied prospectively, as it could affect patient outcomes and surgical decisions. Although OPCAB shows a higher discrepancy, it may not translate to functional limitations, especially with hybrid procedures like PCI and stenting becoming more common [12]. Thus, this difference might not be a major drawback. Furthermore, the proposed benefit of ONCAB of complete revascularization is countered by its higher need for ECLS support.

## 5. Conclusions

Our comprehensive, retrospective analysis revealed that while both OPCAB and ONCAB are viable surgical options for managing coronary artery disease in HFrEF patients, our data indicate a tangible benefit in favor of OPCAB. This technique was associated with a spectrum of improved outcomes, such as lower incidences of renal insufficiency and perivascular disease, along with fewer postoperative complications including re-sternotomy, the need for ECLS therapy, acute kidney injury, delirium, and sepsis. Additionally, OPCAB patients benefited from notably shorter ICU stays, less mechanical ventilation time, and reduced overall hospital duration. These advantages were reinforced through propensity score matching, which particularly highlighting the role of OPCAB in decreasing the incidence of AKI and the need for ECLS therapy.

Crucially, OPCAB also achieved a significant reduction in mortality risk when compared to ONCAB for HFrEF patients. These findings advocate for preferential consideration of OPCAB in coronary artery bypass grafting for this patient group. Nonetheless, the decision to use OPCAB or ONCAB should be carefully tailored to each patient, integrating specific clinical factors, the patient’s health status, and the surgical team’s expertise. Our study serves to enrich the knowledge base from which these critical decisions are made, aiming to steer surgical choice towards enhanced patient outcomes.

The nuanced, retrospective comparison undertaken in our research is not just to validate the efficacy of these surgical techniques, but to guide therapeutic decisions towards improved care quality. It is with this objective in mind that we emphasize the importance of individualized surgical approaches. We concur that while surgeon preference and experience are significant, they should be informed by the current evidence, which we have presented. In doing so, our aim is to catalyze informed, patient-centric surgical decisions and encourage the adoption of practices that our research has shown to be beneficial. To solidify the foundation of these recommendations, we advocate for further prospective multicenter studies to confirm these results and refine guidelines for the optimal use of OPCAB and ONCAB in CABG for HFrEF patients. Clinical Perspectives:

The findings presented in this manuscript contribute valuable insights to the field of cardiac surgery, specifically focusing on the outcomes of patients with heart failure with reduced ejection fraction (HFrEF) undergoing coronary artery bypass grafting (CABG) using two different surgical techniques: off-pump coronary artery bypass (OPCAB) and on-pump coronary artery bypass (ONCAB).

The clinical implications of the study are significant, as they shed light on the potential benefits of OPCAB in HFrEF patients. The manuscript highlights several key clinical competencies and translational outlook implications that arise from these findings:

Patient selection and risk stratification: this study emphasizes the importance of thorough preoperative assessment and risk stratification in patients with HFrEF. The higher preoperative risk scores (STS Score and EuroScore II) in the OPCAB group suggest that careful consideration of patient characteristics is crucial when choosing the optimal surgical technique. Clinicians should assess factors such as NYHA classification, renal function, and comorbidities to identify patients who might benefit most from OPCAB.

Minimizing perioperative complications: this study provides evidence that OPCAB is associated with reduced rates of postoperative complications such as acute kidney injury, delirium, and sepsis. These findings emphasize the potential of OPCAB to contribute to improved perioperative outcomes, shorter mechanical ventilation time, and a reduced length of stay in the intensive care unit. Clinicians should consider these benefits when planning and discussing surgical options with patients.

Individualized treatment approach: this manuscript underscores the importance of individualized treatment approaches based on patient characteristics and surgical expertise. Surgeons and heart teams should carefully evaluate each patient’s clinical profile, including factors like renal function, coronary anatomy, and the urgency of the procedure when deciding between OPCAB and ONCAB. These considerations are essential for optimizing patient outcomes and reducing the risk of complications.

Long-term survival and mortality risk: this study’s multivariable logistic regression analysis highlights the potential for OPCAB to reduce the risk of 30-day mortality compared to ONCAB, particularly in patients with HFrEF. This finding has significant implications for clinical decision making and patient counseling, indicating that OPCAB may offer a survival advantage in this specific patient population.

Incomplete revascularization and future research: this study raises questions about the clinical significance of incomplete revascularization and its impact on long-term outcomes. The discussion calls for further research to define the clinical relevance of complete revascularization and its association with patient survival. Future studies should explore the potential benefits of hybrid procedures involving both surgical and percutaneous interventions to achieve optimal revascularization.

In summary, the clinical perspectives in this manuscript underscore the importance of patient-centered care, risk stratification, and surgical technique selection in HFrEF patients undergoing CABG. The findings suggest that OPCAB may offer advantages in terms of perioperative complications, mechanical ventilation time, and mortality risk reduction. However, the choice between OPCAB and ONCAB should be guided by a comprehensive evaluation of patient characteristics, individualized treatment goals, and the expertise of the surgical team. Further research is needed to validate and expand upon these findings, providing a comprehensive understanding of the optimal surgical approach for HFrEF patients requiring CABG.

## Figures and Tables

**Figure 1 biomedicines-11-03043-f001:**
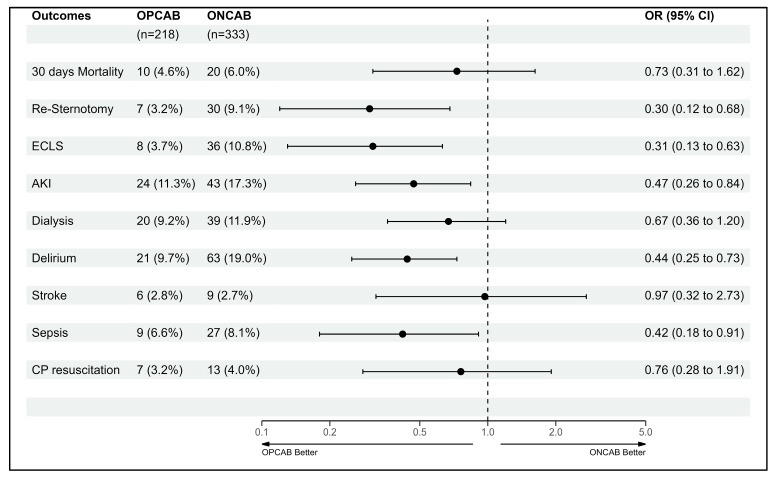
Forestplot total cohort (FP1).

**Figure 2 biomedicines-11-03043-f002:**
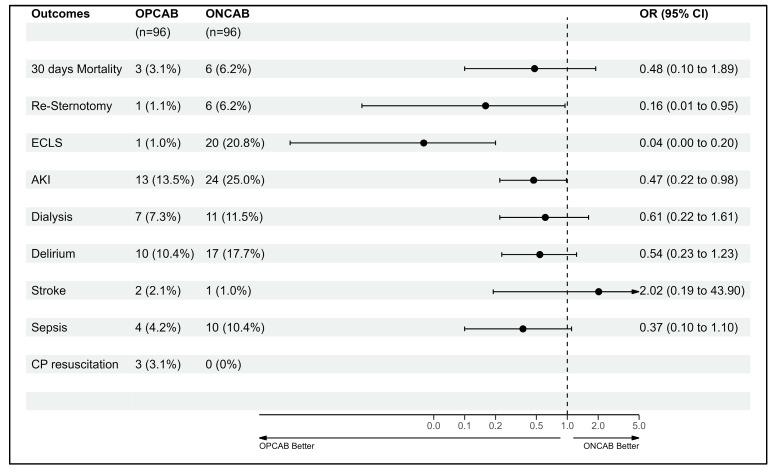
Forestplot after propensity score.

**Table 1 biomedicines-11-03043-t001:** Cohort baseline characteristics.

Variable	Total Cohort (n = 551)	OPCAB (n = 218)	ONCAB (yes = 333)	*p* Value
Gender				
Male n (%)	487 (88.4%)	186 (85.3%)	301 (90.4%)	0.093^Chi2^
Female n (%)	64 (11.6%)	32 (14.7%)	32 (9.6%)	
Age mean (±SD)	67.9 (±9.9)	67.5 (±9.4)	68.1 (±10.2)	0.45^TT^
BMI mean (±SD)	27.9 (±4.9)	28.2 (±5.2)	27.8 (±4.7)	0.38^TT^
Diabetes				
OAD n (%)	127 (23.1%)	54 (24.8%)	73 (22.0%)	0.71^Chi2^
Insulin dependent n (%)	133 (24.2%)	50 (22.9%)	83 (25%)	
Smoking history				
Former n (%)	155 (28.5%)	64 (30.2%)	91 (27.4%)	0.123^Chi2^
Active n (%)	112 (20.6%)	51 (24.0%)	61 (18.4%)	
Hypertension	516 (93.6%)	201 (92.2%)	315 (94.6%)	0.34^Chi2^
COPD	107 (19.4%)	39 (17.9%)	68 (20.4%)	0.53^Chi2^
Hyperlipidemia	467 (85.1%)	174 (80.2%)	293 (88.3%)	0.013^Chi2^
Renal insufficiency				
0 n (%)	384 (70.8%)	138 (65.1%)	246 (74.5%)	0.002^LL^
1 n (%)	59 (10.9%)	19 (9.0%)	40 (12.1%)	
2 n (%)	33 (6.1%)	19 (9.0%)	14 (4.2%)	
3 n (%)	50 (9.2%)	27 (12.7%)	23 (7.0%)	
4 n (%)	8 (1.5%)	6 (2.8%)	2 (0.6%)	
5 n (%)	8 (1.5%)	3 (1.4%)	5 (1.5%)	
Carotid stenosis	93 (16.9%)	39 (17.9%)	54 (16.2%)	0.69^Chi2^
Perivascular disease	127 (23.1%)	65 (29.8%)	62 (18.6%)	0.003^Chi2^
NYHA Class				
1 n (%)	20 (3.6%)	11 (5.0%)	9 (2.7%)	0.002^LL^
2 n (%)	240 (43.6%)	64 (29.4%)	176 (52.8%)	
3 n (%)	241 (43.7%)	117 (53.7%)	124 (37.2%)	
4 n (%)	50 (9.1%)	26 (11.9%)	24 (7.2%)	
LVEF pre-Op mean (±SD)	31.4 (±6.9)	31.5 (±6.8)	31.3 (±7.0)	0.69^TT^
Number of disease vessels median (IQR)	3 (3–3)	3 (3–3)	3 (3–3)	0.25^MW^
Left main affected				
No n (%)	333 (60.4%)	129 (59.2%)	204 (61.3%)	0.69^Chi2^
Yes n (%)	218 (39.6%)	89 (40.8%)	129 (38.7%)	
Prior MI				<0.001^Chi2^
No n (%)	337 (61.1%)	107 (49.1%)	230 (69.1%)	
Yes n (%)	214 (38.9%)	111 (50.9%)	103 (30.9%)	
Aortic stenosis	24 (4.4%)	15 (6.9%)	9 (2.7%)	0.031^Chi2^
Type of surgery				
Elective n (%)	310 (56.3%)	100 (45.9%)	210 (63.1%)	<0.001^Chi2^
Urgent n (%)	157 (28.5%)	87 (40.0%)	70 (21.0%)	
Emergent n (%)	84 (15.2%)	31 (14.1%)	53 (15.9%)	
STS score median (IQR)	1.9 (0.9–3.6)	2.5 (1.1–4.8)	1.6 (0.8–3.1)	<0.001^MW^
EuroScore II median (IQR)	3.6 (2.02–7.03)	3.9 (2.5–7.3)	3.3 (1.9–6.9)	0.03^MW^

Chi2: Chi-Square. MW: Mann-Whitney U test. TT: T-Test. LL: Log Likelihood.

**Table 2 biomedicines-11-03043-t002:** Intra and Postoperative Parameters.

Variable	Total Cohort (n = 551)	OPCAB (n = 218)	ONCAB (n = 333)	*p* Value
Number of anastomoses Median (IQR)	3 (2–3)	3 (2–3)	3 (3–4)	<0.001^MW^
Complete revascularization	472 (86.0%)	169 (77.5%)	303 (91.5%)	<0.001^Chi2^
Intraoperative Transfusions				
Packed red blood cells	113 (20.5%)	18 (8.3%)	95 (28.5%)	<0.001^Chi2^
Platelets	123 (22.3%)	18 (8.3%)	105 (31.5%)	<0.001^Chi2^
Fresh frozen plasma	38 (6.9%)	6 (2.8%)	32 (9.6%)	<0.001^Chi2^
Intraoperative Vasopressor and Inotropic requirements				
Adrenaline median (IQR)	0 (0–0.04)	0 (0–0)	0.02 (0–0.05)	<0.001
Noradrenalin median (IQR)	0.1 (0.06–0.16)	0.11 (0.06–0.17)	0.10 (0.05–0.15)	0.172
Dobutamin median (IQR)	0 (0–0)	0 (0–0)	0 (0–0)	0.124
Milrinon median (IQR)	0 (0–0)	0 (0–0)	0 (0–04)	<0.001
Postoperative Parameters and Complications				
Time hospitalization days median (IQR)	13 (9–18)	10 (8–14.8)	14 (11–20)	<0.001^MW^
UCI length (hours) median (IQR)	72.5 (42–144)	53 (24–116)	93 (48–168)	<0.001^MW^
Mecanical ventilation (hours) median (IQR)	13 (8–24)	7.5 (4.5–15)	16 (12–30)	<0.001^MW^

Chi2: Chi-Square. MW: Mann-Whitney U test.

**Table 3 biomedicines-11-03043-t003:** Logistic regression of postoperative complications.

Variable	Total Cohort (n = 551)	OPCAB (n = 218)	ONCAB (n = 333)	OR *	95% CI	*p* Value
Re-Sternotomy due to bleeding	37 (6.8%)	7 (3.2%)	30 (9.1%)	0.30	0.12–0.68	0.006
ECLS	44 (8%)	8 (3.7%)	36 (10.8%)	0.31	0.13–0.63	0.003
AKI	67 (14.5%)	24 (11.3%)	43 (17.3%)	0.47	0.26–0.84	0.012
Dialysis	59 (10.8%)	20 (9.2%)	39 (11.9%)	0.67	0.36–1.20	0.185
Delirium	84 (15.3%)	21 (9.7%)	63 (19.0%)	0.44	0.25–0.73	0.002
Stroke	15 (2.8%)	6 (2.8%)	9 (2.7%)	0.97	0.32–2.73	0.95
Sepsis	36 (6.6%)	9 (4.1%)	27 (8.1%)	0.42	0.18–0.91	0.036
Cardiopulmonary resuscitation	20 (3.7%)	7 (3.2%)	13 (4.0%)	0.76	0.28–1.91	0.58

* Adjusted by EuroScore II and the OR corresponds to OPCAB (reference ONCAB).

**Table 4 biomedicines-11-03043-t004:** Intra- and postoperative parameters after propensity score matching.

		OPCAB (n = 96)	ONCAB (yes = 96)		*p* Value
Time hospitalization days median (IQR)		10 (8–14)	14 (11–23)		<0.001^MW^
UCI length (hours) median (IQR)		46 (23–94)	96 (68–164)		<0.001^MW^
Mechanical ventilation (hours) median (IQR)		7 (4–15)	16 (12–25)		<0.001^MW^
Number of anastomosis Median (IQR)		3 (2–3)	3 (3–4)		<0.001^MW^
	OPCAB (n = 96)	ONCAB (n = 96)	OR *	95% CI	*p* value
Complete revascularization	75 (78.1%)	85 (89.5%)	0.42	0.17–0.93	0.037
Packed red blood cell transfusion	3 (3.1%)	28 (29.2%)	0.08	0.02–0.23	<0.001
Platelets transfusion	5 (5.2%)	26 (27.1%)	0.15	0.05–0.38	0.002
Fresh frozen plasma transfusion	1 (1.1%)	13 (13.5%)	0.07	0.004–0.35	0.001
Re-Sternotomy due to bleeding	1 (1.1%)	6 (6.2%)	0.16	0.01–0.95	0.09
ECLS Therapy	1 (1.0%)	20 (20.8%)	0.04	0.002–0.20	0.002
AKI	13 (13.5%)	24 (25.0%)	0.47	0.22–0.98	0.04
Dialysis	7 (7.3%)	11 (11.5%)	0.61	0.22–1.61	0.33
Delirium	10 (10.4%)	17 (17.7%)	0.54	0.23–1.23	0.150
Stroke	2 (2.1%)	1 (1.0%)			
Sepsi	4 (4.2%)	10 (10.4%)	0.37	0.10–1.1	0.107
Resuscitation	3 (3.1%)	0			
30-days mortality	3 (3.1%)	6 (6.2%)	0.48	0.10–1.89	0.31

OR: Odds Ratio; *: Adjusted by EuroScore II. MW: Mann-Whitney U test.

**Table 5 biomedicines-11-03043-t005:** Multivariable logistic regression model (binary logistic regression) outcome 30-days mortality to LVEF (without propensity score).

	OR	95% CI	*p* Value	AUC	Nagelker	HL	LL Test Chi2
STS score (log transf)	2.54	1.61–4.21	<0.01	0.72	0.09	0.072	
STS score (log transf)	2.54	1.60–4.24	<0.001	0.72	0.10	0.082	0.138
STS score (log transf) OPCAB	2.62 0.57	1.62–4.45 0.24–1.26	<0.001 0.174	0.74	0.11	0.090	0.165
EuroScore II (log transf)	3.29	2.04–5.47	<0.001	0.74	0.135	0.112	
EuroScore II (log transf) OPCAB	3.29 0.73	2.05–5.47 0.31–1.62	<0.001 0.46	0.74	0.137	0.115	0.450
EuroScore II (log transf) OPCAB LVEF pre-OP	3.17 0.74 0.97	1.95–5.32 0.32–1.63 0.92–1.03	<0.001 0.329 0.467	0.74	0.142	0.119	0.332

HL: Hosmer–Lemeshow test.

## Data Availability

Data are contained within the article and the foundational research data can be made available upon request in compliance with the EU’s General Data Protection Regulation (GDPR). To ensure compliance, we will seek legal counsel in this matter.

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
