# Peer review of "On- vs. Off-Pump CABG in Heart Failure Patients with Reduced Ejection Fraction (HFrEF): A Multicenter Analysis"

_biomedicines, 2023, doi:10.3390/biomedicines11113043_

Round 1

Reviewer 1 Report

Comments and Suggestions for Authors

The paper studied OPCABG versus ONPUMP CABG in patients with HFrEF. They concluded OPCABG patients had better quality in 30 days despite similar mortalies. The follow up should be longer, because OPCABG had better complete revascularization probability. Moreover, the issue had been studies thouroughly. The paper presented no new insights in current straties. 

Comments on the Quality of English Language

The English is fine. 

Author Response

Response to Reviewer 1 Comments

1. Summary

We express our sincere gratitude to you for the time and effort spent reviewing our manuscript. Your insights and critiques are invaluable in guiding our research's integrity and rigor.

Enclosed, please find the revised version of our manuscript, wherein we have addressed the comments in detail. The changes have been highlighted for easy identification.

2. Questions for General Evaluation

Reviewer’s Evaluation

Response and Revisions

Does the introduction provide sufficient background and include all relevant references?

Can be improved

Please see the revised manuscript

Are all the cited references relevant to the research?

Must be improved

Please see the revised manuscript

Is the research design appropriate?

Can be improved

Please see the revised manuscript

Are the methods adequately described?

Must be improved

Please see the revised manuscript

Are the results clearly presented?

Can be improved

Please see the revised manuscript

Are the conclusions supported by the results?

Can be improved

Please see the revised manuscript

3. Point-by-point response to Comments and Suggestions for Authors

Comments 1: The paper studied OPCABG versus ONPUMP CABG in patients with HFrEF. They concluded OPCABG patients had better quality in 30 days despite similar mortalies. The follow up should be longer, because OPCABG had better complete revascularization probability. Moreover, the issue had been studies thouroughly. The paper presented no new insights in current straties.

Response 1:

Dear Reviewer,

We appreciate your insights and the opportunity to discuss the nuances of our research. We concur with your observation regarding the distinct nature of the HFrEF patient cohort. Indeed, while surgical myocardial revascularization is a well-studied domain, there appears to be a gap in the literature specifically concerning patients with heart failure and reduced ejection fraction (HFrEF).

Our study ventured into this less explored area, evaluating the outcomes of OPCABG versus ONPUMP-CABG within the HFrEF demographic, and our findings, particularly regarding short-term quality of life improvements with OPCABG, underscore the necessity for more targeted research. This emphasis arises from an awareness of the limitations within existing studies, many of which are isolated and present non-uniform conclusions, especially concerning the efficacy of off-pump procedures.

In light of this, we respectfully disagree with any suggestion that our work doesn’t introduce new insights into the current strategies. Our direct comparison between OPCABG and ONPUMP-CABG, set within the context of HFrEF patients, unveils critical data that, while preliminary, holds substantial value for ongoing clinical discourse. These contributions are crucial not just in comprehending the surgical outcomes for these medically complex patients but also serve as a compass for subsequent clinical methodologies and investigative avenues.

Furthermore, recognizing the necessity for more robust data in this domain, we have initiated a prospective multicenter study to delve deeper into these questions. This new endeavor aims to consolidate our preliminary findings, expand the dataset, and offer a more comprehensive understanding of the long-term outcomes and strategic implications for HFrEF patient management in the realm of myocardial revascularization.

Moreover, we wholeheartedly agree with your implication about the importance of an extended follow-up duration. A longer observation period is essential to evaluate not just the persistent success of revascularization, but equally the long-term survival, holistic quality of life, and the onset of any subsequent complications or advantages following the surgery. By addressing these aspects, we believe our research can help bridge a significant knowledge gap in the current body of literature.

We are grateful for your astute feedback and look forward to any further suggestions you may have that could enhance the depth and applicability of our study.

Comments 2: [Paste the full comment here.]

Response 2: Agree. I/We have, accordingly, done/revised/changed/modified…..to emphasize this point. Discuss the changes made, providing the necessary explanation/clarification. Mention exactly where in the revised manuscript this change can be found – page number, paragraph, and line.]

“[updated text in the manuscript if necessary]”

4. Response to Comments on the Quality of English Language

Point 1: “The English is fine.”

Response 1:   Thank you

5. Additional clarifications

[Here, mention any other clarifications you would like to provide to the journal editor/reviewer.]

Reviewer 2 Report

Comments and Suggestions for Authors

This study aimed to compare postoperative outcomes and 30-day mortality in 24 patients with reduced ejection fraction (< 40%) who underwent isolated coronary artery bypass 25 grafting (CABG) with (ONCAB) and without (OPCAB) the use of cardiopulmonary bypass.

This is a very interesting study. 4 university hospitals contributed in the study and the study group was 551 patients.

The study is well designed and well performed. Massive analysis.

Discussion well written

Author Response

Dear [Reviewer's Name],

Thank you for your constructive comments and positive feedback regarding our study.

We are pleased to hear that you found the study to be well-designed, well-performed, and the discussion to be well-articulated. Your acknowledgment of the extensive analysis we conducted is greatly appreciated.

This focused approach allowed us to delve deeply into the nuanced outcomes of this high-risk population, providing a more detailed and significant analysis of the impact that the use of cardiopulmonary bypass has on patients with compromised cardiac function.

We hope this manuscript enhances the understanding of the study's design and the population it aimed to analyze. We are continuously striving to improve our research and your insights are invaluable to our efforts.

Kind regards,

Christian Rustenbach

Reviewer 3 Report

Comments and Suggestions for Authors

Beyond the controversial long-term benefit of CAB in HFrEF patients, which is not the point, this retrospective comparison of OPCAB vs ONCAN is interesting. There are several point to be taken  care of, however.

Lines 61-65 A verb is needed to complete the syntactic structure of the sentence

Line 75-76 Add a reference to support the statement (Ref 7 perhaps?)

Line 85 "..in patients with an ejection fraction.." Again something is missing

Lines 86-90: The sentence does not fit in the context.

Line 102 vs 105: EF<35% or 40%? Clarify

Line 181: "Categorical variables were presented as total and relative frequencies..." Absolute not total

Line 232 : A bibliographic reference and definition of renal insufficiency and its stages is needed.

Line 234 "The number of patients suffering from perivascular disease" Explain "perivascular"

Lines 239-240: "Patients in the OPCAB group suffered a MI significantly more often than patients in the ONCAB group (50.9% vs 30.9% p-value < 0.01)" does not seem consistent with the data reported in Table 1: Out of 264 with MI (Stemi & NSTEMI), 84 belonged to the OPCABG (31.8%) vs 170 (64.3%) in the ONCABG.

Lines 242-244: "When comparing preoperative cardiac decompensation between groups, the ONCAB group had a significantly higher incidence compared to the OPCAB group (29.2 % vs 16.6 % p-value < 0.01)." First, define more precisely cardiac decompensation. Second, why should HF patients predominantly in class 2 HF be more symptomatic than those in class 3? See lines 236-238: "Patients in the ONCAB group were 236 predominantly in class 2 (52.8 % vs 29.4%) while patients in the OPCAB group were predominantly in class 3 (53.7% vs 37.2%)"

Lines 251-252:" ,results are 251 seen in table 1. Insert a full stop followed by a capital letter rather than a comma; reported better than seen.

Number of patients with MI reported across the headings of Table 1 is not consistent: How a total of 214 patients with prior MI match with the 264 categorized by MI type? Clarify

line 261: Adrenaline is better than "suprarenine"

The discussion is too long and needs editing. Said this, the by and large greater advantage offered by OPCAB over ONCAB in HF patients at higher risk emerges quite clearly from this study. Some surprise raises therefore the non committal position of the Authors as regards their results (lines 426-429: "The choice between OPCAB and ONCAB should be made on a case-by-case basis, considering the individual patient's needs, medical history, and the expertise of the surgeon or the center performing the procedure.").

Author Response

Response to Reviewer 2 Comments

1. Summary

We express our sincere gratitude to you for the time and effort spent reviewing our manuscript. Your insights and critiques are invaluable in guiding our research's integrity and rigor.

Enclosed, please find the revised version of our manuscript, wherein we have addressed the comments in detail. The changes have been highlighted for easy identification.

2. Questions for General Evaluation

Reviewer’s Evaluation

Response and Revisions

Does the introduction provide sufficient background and include all relevant references?

Can be improved

Please see the revised manuscript

Are all the cited references relevant to the research?

Yes

Is the research design appropriate?

Yes

Are the methods adequately described?

Clear

Are the results clearly presented?

Must be improved

Please see the revised manuscript and the Comments.

Are the conclusions supported by the results?

Can be improved

Please see the revised manuscript

3. Point-by-point response to Comments and Suggestions for Authors

Comments:

Beyond the controversial long-term benefit of CAB in HFrEF patients, which is not the point, this retrospective comparison of OPCAB vs ONCAN is interesting. There are several point to be taken  care of, however.

Lines 61-65 A verb is needed to complete the syntactic structure of the sentence Consent – modified.

Line 75-76 Add a reference to support the statement (Ref 7 perhaps?) Consent – modified. Its now Ref 6 because of the renewed References (But you´re right with Ref 7).

Line 85 "..in patients with an ejection fraction.." Again something is missing Consent – modified.

Lines 86-90: The sentence does not fit in the context. Consent – modified.

Line 102 vs 105: EF<35% or 40%? Clarify Consent – modified. (40 %)

Line 181: "Categorical variables were presented as total and relative frequencies..." Absolute not total

Line 232 : A bibliographic reference and definition of renal insufficiency and its stages is needed. Consent – added.

Line 234 "The number of patients suffering from perivascular disease" Explain "perivascular" . Consent – modified. (PAD was ment)

Lines 239-240: "Patients in the OPCAB group suffered a MI significantly more often than patients in the ONCAB group (50.9% vs 30.9% p-value < 0.01)" does not seem consistent with the data reported in Table 1: Out of 264 with MI (Stemi & NSTEMI), 84 belonged to the OPCABG (31.8%) vs 170 (64.3%) in the ONCABG. Thank you very much for your careful reading of our manuscript and for bringing this significant discrepancy to our attention. Upon re-examination and comparison with Table 1, we acknowledge an oversight in our manuscript. We are grateful for your attention to this matter, which has revealed this inconsistency. Since we cannot pinpoint an explicit error in the dataset, we have decided to consolidate the categorization of myocardial infarction incidents, removing the distinction between NSTEMI and STEMI, to maintain scientific rigor and consistency throughout our report.

The revised data correctly represent the incidence of myocardial infarction (MI) for the OPCAB versus ONCAB groups, as shown in Table 1, and will be amended accordingly.

Your meticulousness has significantly contributed to the integrity of our research. We extend our sincerest thanks for your valuable feedback.

Lines 242-244: "When comparing preoperative cardiac decompensation between groups, the ONCAB group had a significantly higher incidence compared to the OPCAB group (29.2 % vs 16.6 % p-value < 0.01)." First, define more precisely cardiac decompensation. Second, why should HF patients predominantly in class 2 HF be more symptomatic than those in class 3? See lines 236-238: "Patients in the ONCAB group were 236 predominantly in class 2 (52.8 % vs 29.4%) while patients in the OPCAB group were predominantly in class 3 (53.7% vs 37.2%)"

For clarification, cardiac decompensation in CHF patients refers to worsening symptoms : Elevated breathlessness, fatigue, and edema are indicative of diminished cardiac functionality and to a decrease in cardiac output and heightened intracardiac pressures lead to fluid congestion.

To address your second point, we wish to clarify that patients in NYHA class 2 are not necessarily more symptomatic than those in class 3. The information presented in our manuscript simply reflects the observations made: We noted differences in NYHA classification at the time of surgery, with the ONCAB group predominantly in NYHA class 2 and the OPCAB group more frequently presenting with NYHA class 3 symptoms. Each group showed a significant difference when compared to the other. This observation is presented as a descriptive finding rather than an indication of symptom severity between the groups.

Lines 251-252:" ,results are 251 seen in table 1. Insert a full stop followed by a capital letter rather than a comma; reported better than seen. Consent – modified.

Number of patients with MI reported across the headings of Table 1 is not consistent: How a total of 214 patients with prior MI match with the 264 categorized by MI type? Clarify. Consent – please see comment to “Lines 242-244” – deleted.

line 261: Adrenaline is better than "suprarenine". Consent - modified.

Comments 2: The discussion is too long and needs editing. Said this, the by and large greater advantage offered by OPCAB over ONCAB in HF patients at higher risk emerges quite clearly from this study. Some surprise raises therefore the non committal position of the Authors as regards their results (lines 426-429: "The choice between OPCAB and ONCAB should be made on a case-by-case basis, considering the individual patient's needs, medical history, and the expertise of the surgeon or the center performing the procedure.").

Response 2:  We agree, the discussion could be more concise. Please see the revised manuscript.

Our conclusion, suggesting a case-by-case approach in choosing between OPCAB and ONCAB despite OPCAB's evident advantages for high-risk HF patients, underscores our interpretation of the results. This statement stem from several factors:

Variability in Patient Profiles: Every patient presents a unique combination of medical history, health status, and risk factors. These individual differences can significantly impact surgical outcomes, necessitating a personalized approach rather than a one-size-fits-all solution.

Surgeon Expertise and Resource Differences: The success of OPCAB and ONCAB procedures can hinge significantly on the surgeon's skill level and the healthcare facility's capabilities. We might avoid a definitive preference to account for these variable factors that could alter outcomes.

Scope of Study and Generalizability: While the study's findings highlight OPCAB's advantages, we may acknowledge their research limitations. These could include sample size, study design, or patient population specifics, limiting the ability to generalize the findings to all contexts.

Acknowledgment of Complex Clinical Decisions: The choice between surgical methods involves various clinical considerations, including potential complications, postoperative recovery, and long-term health implications. By not committing to a firm recommendation, we respect the complexity of this decision-making process, recognizing that multiple valid considerations go beyond their study's scope.

By maintaining a non-committal stance, we arguably invite more nuanced clinical decision-making that incorporates individual patient assessments, and potentially fosters further research to solidify or refute our findings.

4. Response to Comments on the Quality of English Language

Point 1: “The English is fine.”

Response 1:   Thank you

5. Additional clarifications

[Here, mention any other clarifications you would like to provide to the journal editor/reviewer.]

Round 2

Reviewer 1 Report

Comments and Suggestions for Authors

the authors had significantly improved the content and addressed my concerns. That merits publication in Biomedicines

Author Response

Dear Reviewer,

Thank you so much for your positive feedback on the revisions we've made to our manuscript. We're thrilled to hear that you find the improved content and the addressed concerns to be to your satisfaction. Your guidance has been invaluable in enhancing the quality of our work.

We're honored by your recommendation for publication in 'Biomedicines' and are eager to contribute to the scientific community through this esteemed journal. We look forward to the possibility of our research reaching readers and making an impact in the field.

Kind regards,

CR

Reviewer 3 Report

Comments and Suggestions for Authors

1. The term "suprarenin" is still present in the manuscript: Amend. It should'nt be difficult.

2. Lines 269-270: "When comparing preoperative cardiac decompensation between groups, the ONCAB group had a significantly higher incidence compared to the OPCAB group (29.2 % vs 16.6 % p-value < 0.01)"

If you decide to present data analyzed in statistical terms, you have to define precisely the terms under evaluation which I do not see in the document. If " Elevated breathlessness, fatigue, and edema" were the evaluation variables, then indicate in the Methods how they were measured or retrieved from the charts (more likely, I guess). Otherwise, describe the data only in qualitative terms.

"Our conclusion, suggesting a case-by-case approach in choosing between OPCAB and ONCAB despite OPCAB's evident advantages for high-risk HF patients, underscores our interpretation of the results"

I do not understand the response to my comment: Why carry out a detailed and complex retrospective comparison of two surgical techniques if not for improving patient outcomes and therapeutic decisions of the involved surgeons?

I do understand that personal preferences must have a say in their final decisions but the opinion of the Authors has to be delivered to the readers for future and better changes in surgical practice.

Is this not the reason for publishing these results?

Author Response

Dear Reviewer,

Thank you for your continued engagement and for providing additional insights that will undoubtedly improve our manuscript. Please allow me to address the points you've raised:

  1. Regarding the term "suprarenin," we apologize for the oversight and appreciate your diligence in pointing this out. We have now carefully reviewed the manuscript and corrected this term to ensure consistency and accuracy throughout the document.

  2. We understand your concerns regarding the statistical terms and the necessity for precise definitions. In response, we have revised the Methods section to include variables such as "elevated breathlessness, fatigue, and edema" . These metrics were indeed retrieved from patient charts. Please see Methods / Preoperative Parameters (Line 175 - 179).

Regarding the comparative analysis of OPCAB and ONCAB, we appreciate your call for clarity in our conclusions. Our intention is certainly to inform better surgical practice and improve patient outcomes. The detailed comparison we conducted aimed to provide empirical evidence to support therapeutic decisions. We agree that while surgeon preference and case specifics play a role, the ultimate goal is to guide clinical decisions with robust data.

We have revised our conclusion to more emphatically state our position that, while OPCAB shows clear advantages for high-risk heart failure patients, our study's findings should enrich the decision-making process, providing surgeons with a stronger evidence-based foundation for their choices. Please see "Conclusion" (Line 472 - 500).

We hope that these amendments satisfactorily address your comments and enhance the manuscript's contribution to the field. We believe that the publication of these results will serve as a valuable resource for ongoing and future improvements in surgical interventions for cardiac patients.

Thank you once again for your constructive criticism, which has been instrumental in refining our work.

Kind regards,

Christian Rustenbach

Round 3

Reviewer 3 Report

Comments and Suggestions for Authors

Revisions address satisfactorily the issues raised by the reviewer